



# CHROTRAN 1.0: A mathematical and computational model for in situ heavy metal remediation in heterogeneous aquifers

Scott K. Hansen[1], Sachin Pandey[1], Satish Karra[1], and Velimir V. Vesselinov[1]

[1]Computational Earth Science Group (EES-16), Los Alamos National Laboratory

*Correspondence to:* Scott K. Hansen (skh3@lanl.gov)

**Abstract.** Groundwater contamination by heavy metals is a critical environmental problem for which in situ remediation is frequently the only viable treatment option. For such interventions, a three-dimensional reactive transport model of relevant biogeochemical processes is invaluable. To this end, we developed a model, CHROTRAN, for in situ treatment, which includes full dynamics for five species: a heavy metal to be remediated, an electron donor, biomass, a nontoxic conservative bio-inhibitor, and a biocide. Direct abiotic reduction by donor-metal interaction as well as donor-driven biomass growth and bio-reduction are modeled, along with crucial processes such as donor sorption, bio-fouling and biomass death. Our software implementation handles heterogeneous flow fields, arbitrarily many chemical species and amendment injection points, and features full coupling between flow and reactive transport. We describe installation and usage and present two example simulations demonstrating its unique capabilities. One simulation suggests an unorthodox approach to remediation of Cr(VI) contamination.



## 1 Introduction

Heavy metals, including chromium, arsenic, copper, nickel, selenium, technetium, uranium, and zinc, are widespread and hazardous subsurface contaminants in groundwater aquifers (Appelo and Postma, 2004; Tchounwou et al., 2012). For many heavy metals, their most stable oxidation state is often the most toxic (Duruibe et al., 2007; Hashim et al., 2011), and this

oxidation state is typically the highest that occurs under near-surface conditions. Additionally, the chemical reduction of certain metals is known to reduce their mobility (Violante et al., 2010). This has inspired efforts to manipulate in situ conditions to stimulate microbial growth and achieve biologically mediated metals reduction. This technique has been demonstrated, at least in some settings, for chromium, uranium and selenium (Lovley, 1993, 1995), nickel (Zhan et al., 2012), technetium (Istok et al., 2004), and copper (Andreazza et al., 2010), and has been noted as a viable bioremediation technique by recent critical

reviews (Hashim et al., 2011; Wu et al., 2010). Bioprecipitation, a process by which microbiological exudates react with metals to produce an insoluble compound, has been widely observed (Malik, 2004; Van Roy et al., 2006; Radhika et al., 2006) and has been noted by Wu et al. (2010) as a remediation method. Bio-stimulants have also recently been shown to effectively reduce chromium through abiotic redox pathways (Chen et al., 2015; Hansen et al., 2016), and after fermentation for other metals (Hashim et al., 2011). Naturally, designing a remedial intervention using one of this family of techniques benefits

greatly from the use of a multi-dimensional/multi-component numerical model of groundwater flow, contaminant transport, and biogeochemical processes to evaluate different remediation strategies under varying field conditions. The model should be capable of capturing the transport behavior of electron donors, biomass, and other species, dominant biogeochemical reactions, and how these processes influence and are influenced by subsurface flow.

Although the development on in situ bioreactive transport models goes back to at least the 1980s, the literature is not vast.

Early work focused on in situ bioremediation of toxic organic compounds through oxidation. A thorough mathematical and 2D numerical study representative of this approach is due to Chiang et al. (1991), who presented a three-equation model involving a mobile electron donor (assumed to be the contaminant), mobile dissolved oxygen, and immobile biomass. The contaminant was assumed to be consumed only in the microbial growth reaction, which was linear in biomass, Monod (Monod, 1949) in electron donor, and Monod in electron acceptor. Wheeler et al. (1992) subsequently extended a reactive model of this sort to three

dimensions to simulate biodegradation of $CH_4$. Travis (1993) presented a more complicated, unsaturated three-dimensional model, which introduced Monod dependence on nutrients, and the potential for two electron donors, with one inhibiting the other. This approach was further elaborated upon in a study of TCE degradation (Travis and Rosenberg, 1998) by accounting for living and dead microbes, microbial predators, and first-order kinetic sorption of all aqueous species (microbes were treated as mobile). Another complex oxidation model was developed by Suk et al. (2000), which explicitly modeled both mobile and

immobile biomass, contained a decay network, and featured both anaerobic and aerobic oxidation, in competition.

The development of models for metal reduction is comparatively more recent. For U(VI), field-scale modeling studies have been performed on bio-reduction under anaerobic conditions at the Old Rifle Site in Colorado (Li et al., 2010, 2011; Yabusaki et al., 2011). These conceptions treat the contaminant as the sole electron acceptor, with an externally applied electron donor,



and the implied equations have a similar form to those devised by Chiang et al. (1991): linear in biomass, Monod in contaminant, and Monod in electron donor. For clarity, this is expressed symbolically as:

$$\frac{\partial C}{\partial t} \propto \frac{\partial D}{\partial t} \propto B \frac{C}{K_C + C} \frac{D}{K_D + D}, \qquad (1)$$

where $C$, is the U(VI) concentration, $B$ is the biomass concentration, $D$ is the electron donor concentration, $K_C$ is the metal reduction Monod constant, and $K_D$ is the electron donor Monod constant. $K_C$ and $K_D$ respectively represent the concentration of $C$ and $D$ at which the reaction rate is halved. Recently, Molins et al. (2015) have published a numerical study of a column experiment with multiple species, all of whose dynamics are of the above form, but including an extra chemical inhibition factor. The models of Li et al. (2010, 2011) were implemented at field-scale in CrunchFlow (Steefel et al., 2015), using its capability to represent single and multiple Monod formulations.

Systems of governing reactive transport equations for enzymatic microbial Cr(VI) reduction have been presented by Alam (2004), and by Shashidhar et al. (2007). Shashidhar et al. (2007) described the Cr(VI) degradation reaction slightly differently from Li et al. (2011):

$$\frac{\partial C}{\partial t} \propto \frac{\partial D}{\partial t} \propto B \frac{K_{C'}}{K_{C'} + C} \frac{D}{K_D + D}. \qquad (2)$$

$K_{C'}$ is the concentration of Cr(VI) at which the reaction rate is halved, which is similar to $K_C$. However, although (2) appears

superficially similar to (1), the $C$ factor represents entirely different behavior: not as an energy source but rather as an inhibitor. Interestingly, since the RHS of (2) is a proxy for the biomass growth reaction, $C$ consumption is modeled as proportional to biomass growth, but the biomass growth rate is modeled as independent of $C$. Biomass dynamics were governed by a growth term proportional to donor consumption and a first-order decay term, accounting for eventual biomass die-off. Other authors (e.g., Somasundaram et al., 2009) have used a similar approach. Alam (2004) presented a relatively complex model

which included transport with both mobile and immobile biomass, and also included two enzymes (both created due to biomass growth, but one conserved, and one irreversibly consumed during bio-reduction). Neglecting the irreversibly-consumed enzyme and the mobile-immobile behavior, this model shares its electron donor and biomass dynamics with the model of Shashidhar et al. (2007). It differs significantly from other models that we are aware of by treating the Cr(VI) degradation reaction in this model as an incidental enzymatic process, and is governed by the following Monod equation:

$$\frac{\partial C}{\partial t} \propto B \frac{C}{K_C + C}. \qquad (3)$$

There is strong experimental support for this approach (e.g., Okeke, 2008), and this is arguably more defensible in a real complex geochemical system in which there are multiple competing donors and receptors, and given that there is evidence for indirect reduction pathways, e.g., by metabolites (Priester et al., 2006). All of the models of Cr(VI) bio-reduction, discussed above, appear to be one-dimensional only.

Our literature review did not reveal discussion of field-scale bio-reduction models for other heavy metal species. It thus appears that the primary example of a bio-reduction model applicable to modeling a real-world remediation scheme is the CrunchFlow model of uranium treatment at the Rifle site, which was discussed above. We set out to develop a new model, dubbed CHRO-



TRAN, which is optimized for modeling bioremediation of Cr(VI), but of sufficient generality that it might be used for bioremediation of other metals, or for abiotic reduction, with ease. The key features of the model we developed are as follow:

**Direct abiotic reaction between electron donor and contaminant**  Recent experimental results (Chen et al., 2015; Hansen et al., 2016) have established a rapid direct redox reaction when molasses is used as an electron donor and Cr(VI) is the contaminant, rather than the bio-mediated reaction previously posited. It is thus crucial to include this behavior in a model aimed at remediation design.

**Indirect Monod kinetics**  On account of the evidence (Wang and Xiao, 1995; Okeke, 2008; Hansen et al., 2016) for modeling Cr(VI) degradation with (3), we implemented this general formulation as opposed to one which ties all contaminant degradation to a single biomass growth equation.

**Bio-fouling / Bio-clogging**  It is well known in practice that one of the problems afflicting bioremediation schemes is build-up of biological material near the amendment injection point. This reduces the hydraulic conductivity, interfering with amendment injection, and may rapidly consume any amendment that does manage to pass through it. The model thus contains feedback between local biomass concentration and flow parameters such as porosity and hydraulic conductivity.

**Biomass crowding**  Similarly, if biomass becomes overly dense, this causes cell stress, which reduces the rate of further growth. Since clogging is enabled, this behavior was added as well.

**Modeling of amendment additives**  To address clogging or to attempt to spread electron donors farther from the well before they are consumed, additional chemicals may be injected to reduce biomass concentrations, and their reactive transport behavior is incorporated.

**Multiple donor consumption pathways**  The best model of electron donor consumption by biomass may be proportional to biomass concentration or biomass growth, and the model can handle any such combination.

Building this functionality required custom programming beyond what is embedded in existing reactive transport codes (Steefel et al., 2015). To accomplish our goal, we turned to PFLOTRAN (Lichtner et al., 2015), which is open source, has a modular structure, and a "reaction sandbox" interface (Hammond, 2015) that allows derivative versions with custom reaction behavior to be developed and compiled. In this fashion, no changes to the flow and transport part of PFLOTRAN are needed. We developed CHROTRAN based on the existing PFLOTRAN code framework, taking advantage of the reaction sandbox interface to implement complex model features not included in basic microbial packages and leverage other aspects of PFLOTRAN, such as its high-performance computing capabilities.

In Section 2, we present the mathematical details of CHROTRAN and justify some of the decisions underlying the model. In Section 3, we present two numerical studies which illustrate CHROTRAN and also suggest an interesting conclusion regarding Cr(VI) remediation. In Section 4, we briefly summarize what has been presented. The CHROTRAN v. 1.0 user manual is presented in Appendix A, which gives instructions on how to install and use the software.



## 2 Model description

We consider flow and transport at aquifer scale. Conceptually, the aquifer is modeled as saturated, with incompressible water moving in accordance with Darcy's law (CHROTRAN can also simulate partially-saturated vadose-zone flow and transport). Two transport processes are considered, namely, advection with the Darcy flow and Fickian dispersion. Multiple reaction terms are then added in order to capture the complex chemical dynamics during remediation. As the model is intended to be used for remedial design, every effort was made to simplify the formulation to use the smallest number of explanatory variables and parameters, and to keep the equations at a high level of abstraction, so they are not tied to one particular set of chemical species.

The following are the several species whose dynamics are captured by the system of reaction equations, each with their own symbols:

**Biomass,** $B$ $[\mathrm{mol}\,\mathrm{L}_\mathrm{b}^{-1}]$**,** representing the concentration of all microbes and their associated extracellular material. The quantification of biomass as a "molar" rather than a mass concentration is unusual, and was done for two reasons: (i) to avoid hard-coding units in which biomass concentration is to be specified, and (ii) to simplify presentation of the model, so all governing equations have the same units. A mole of biomass should be understood as an equivalent mass: any quantity can be used, as long as one uses a consistent definition throughout the model. In the examples in this paper, we use the definition 1 mol ≡ 1 g of biomass.

**Aqueous contaminant,** $C$ $[\mathrm{mol}\,\mathrm{L}^{-1}]$**,** which we here assume is a heavy metal ion in its oxidized state, such as Cr(VI) or U(VI).

**Electron donor,** which is part of the chemical amendment, and may be

1. immobile, represented by $D_i$ $[\mathrm{mol}\,\mathrm{L}_\mathrm{b}^{-1}]$, or

2. mobile, represented by $D_m$ $[\mathrm{mol}\,\mathrm{L}^{-1}]$,

with exchange of mass between the two states.

**Nonlethal biomass-growth inhibitor,** $I$ $[\mathrm{mol}\,\mathrm{L}^{-1}]$**,** such as ethanol, which is modeled as a conservative species but acts to slow microbial growth.

**Biocide,** $X$ $[\mathrm{mol}\,\mathrm{L}^{-1}]$**,** which reacts directly with biomass and is consumed.

For convenience, we also define a total species aqueous concentration of the electron donor, $D$, according to the formula $D = \frac{D_i}{\theta(\boldsymbol{x},t)} + D_m$ $[\mathrm{mol}\,\mathrm{L}^{-1}]$, where $\theta(\boldsymbol{x},t)$ [-] is the current porosity at $\boldsymbol{x}$. For simplicity, we assume that both the mobile and immobile donor participate equally in all reactions.





### 2.1 Flow and transport

#### 2.1.1 Groundwater flow equations

Flow may be modeled using the balance of water mass given by

$$\frac{d}{dt}(\rho_w \theta) + \nabla \cdot \boldsymbol{q} = q_M(\boldsymbol{x}, t), \tag{4}$$

with the water mass fluxes related to head via Darcy's law:

$$\boldsymbol{q} = -\nabla(\rho_w K(\boldsymbol{x}, \boldsymbol{t}) h(\boldsymbol{x}, \boldsymbol{t})), \tag{5}$$

where $K(\boldsymbol{x})$ [m/s] is the local hydraulic conductivity and $h(\boldsymbol{x}, t)$ [m] is the local hydraulic head, $\theta$ is the porosity and $q_M(\boldsymbol{x})$ [kg m$^{-3}$ s$^{-1}$] is the local mass injection rate into the system and $\rho_w$ [kg m$^{-3}$] is the density of water. We note that, since CHROTRAN is built on top of PFLOTRAN, it inherits all of PFLOTRAN's groundwater flow modeling capabilities. This includes the ability to consider unsaturated and otherwise multiphase flow conditions, which are out-of-scope for the present discussion. Please see the PFLOTRAN user manual (Lichtner et al., 2015) for details on its complete capabilities.

The hydraulic conductivity is continually updated in accord with the relation

$$K(\boldsymbol{x}, t) = K(\boldsymbol{x}, 0) \frac{\theta(\boldsymbol{x}, t)}{\theta_0}, \tag{6}$$

where $\theta_0$ [-] is the spatially-uniform initial porosity, and $\theta(\boldsymbol{x}, t)$ is calculated according to

$$\theta(\boldsymbol{x}, t) = \theta_0 - \frac{B(\boldsymbol{x}, t)}{\rho_B}, \tag{7}$$

where $\rho_B$ [mol L$^{-1}$] is the intrinsic biomass density. (Note that, using our proposed definition of 1 mol of biomass as 1 g of biomass, 1 mol L$^{-1}$ = 1 kg m$^{-3}$.)

#### 2.1.2 Advective-dispersive transport operator

We define $\mathcal{T}\{\cdot\}$ to be an advective-dispersive transport operator, which characterizes the hydrodynamic effects on solute transport. For $c$, the concentration of an arbitrary mobile species,

$$\mathcal{T}\{c\} \equiv -\boldsymbol{q} \cdot \nabla c + \nabla \cdot (\theta \boldsymbol{D}(\boldsymbol{q}) \nabla c), \tag{8}$$

where $\boldsymbol{D}$ is a dispersion tensor that depends on the longitudinal and transverse dispersivities, molecular diffusion as well as the Darcy flux. For the work in this paper, we will only consider molecular diffusion and thereby we set $\boldsymbol{D} = D_m \boldsymbol{I}$. Note that, while this is not shown explicitly for compactness, all symbols in this equation are functions of $\boldsymbol{x}$ and $t$.



## 2.2 Biogeochemical reactions

We define one governing equation for each species, mobile or immobile, as well as two equations defining reaction rate expressions for algebraic convenience. The equations involve numerous parameters, whose symbols, units, and long-form name in the CHROTRAN input file are summarized in Table 2. The parameter symbols follow a scheme in which the first letter encodes the physical interpretation of the parameter and the subscript specifies the governing equation in which they participate. A symbol beginning with $\Gamma$ is a second-order mass action rate constant, with units [L mol$^{-1}$ s]. A symbol beginning with $K$ is Monod or inhibition constant with units of concentration, mol L$_b^{-1}$ or mol L$^{-1}$, and represents the concentration at which a process rate becomes 50% of its maximum rate, all other parameters being equal. A symbol beginning with $\lambda$ has units of s$^{-1}$ and is interpreted as a pure first-order reaction rate constant. A symbol beginning with $S$ is dimensionless, and represents a stoichiometric relationship between a reaction rate and the consumption rate of a certain species. Before presenting the equations, it is useful to review all of the chemical processes that are incorporated into the model:

**Abiotic reduction** This is an aqueous-phase bimolecular reaction between the electron donor, $D$, and the contaminant, $C$. It is modeled with a classical second-order mass action rate law.

**Bio-reduction** This represents the removal of the contaminant, $C$ by the biomass, $B$. The process is linear in $B$, and Monod in $C$, which is a common assumption for bio-mediated processes. Note that we are not assuming that reduction of $C$ is directly tied to any particular cell metabolic process. This form is sufficiently general that it can capture other bio-remediation processes besides bio-reduction of heavy metals.

**Biocide reaction** This is an inter-phase bimolecular reaction between the biocide, $X$, and the biomass, $B$. It is modeled with a classical second-order mass action rate law, with the added condition that $B$ cannot fall below a specified minimum concentration $B_{min}$.

**Biomass growth** The core biomass growth reaction irreversibly consumes electron donor, $D$ to increase biomass, $B$. As a biologically catalyzed reaction, it is assumed to be linear in $B$ and Monod in $C$. Two inhibition effects are assumed: a biomass crowding term, tunable with exponent $\alpha$, attenuates growth rate as the biomass concentration rises. The nonlethal inhibitor concentration, $I$, also reduces the reaction rate as its concentration increases.

**Mobile-immobile mass transfer (MIMT)** This is a process with first-order kinetics, which models sorptive retardation of the electron donor.

**Natural decay** This is an empirical process reflecting the idea that, if left unstimulated, both the amount of living cells and the amount of extracellular material in the aquifer will ultimately return to their natural background level (i.e. $B_{min}$). This is modeled as a first-order process.

**Respiration** This represents consumption of the electron donor for purposes of life maintenance, unrelated to biomass growth. This is described by a first-order rate law which is proportional to biomass concentration, $B$.



The explanations of the operative processes and of parameter interpretation above help the descriptions of factors and terms in the governing equations presented below.

## 2.3 Reactive transport equations

### 2.3.1 Definitions of convenience reaction variables

5 The biomass growth reaction is linear in biomass concentration, has a Monod dependency on electron donor, a tunable inhibition factor due to biomass crowding, and a classic inhibition factor describing the impact of the nonlethal growth inhibitor (as indicated by comment braces):

$$
\mu_B = \lambda_{B_1} B \overbrace{\frac{D}{K_D + D}}^{e^- \text{donor}} \underbrace{\left( \frac{K_B}{K_B + B} \right)^{\alpha}}_{\text{crowding}} \overbrace{\frac{K_I}{K_I + I}}^{\text{inhibition}} \qquad \left[ \frac{\text{mol}}{\text{L}_b \text{ s}} \right]. \tag{9}
$$

The direct, abiotic reduction reaction is represented by a classic, second-order mass action law:

10 $$
\mu_{CD} = \Gamma_{CD} CD \qquad \left[ \frac{\text{mol}}{\text{L}_b \text{ s}} \right]. \tag{10}
$$

### 2.3.2 Partial differential equations for mobile chemical components

The mobile components are all governed by the advection-dispersion operator, $\mathcal{T}$, defined previously, and also affected by extra terms implementing the chemical processes outlined earlier (as indicated by comment braces):





$$\frac{\partial \theta C}{\partial t} = \mathcal{T}\{C\} - \overbrace{\lambda_C B \frac{C}{K_C + C}}^{\text{bio-reduction}} - \underbrace{S_C \mu_{CD}}_{\text{abiotic reduction}} \qquad \left[\frac{\text{mol}}{\text{L}_\text{b}\ \text{s}}\right], \tag{11}$$

$$\frac{\partial \theta D_m}{\partial t} = \mathcal{T}\{D_m\} - \overbrace{S_{D_1} \frac{D_m}{D} \mu_B}^{\text{biomass growth}} - \underbrace{\lambda_D \frac{D_m}{D} B}_{\text{respiration}} - \overbrace{S_{D_2} \mu_{CD} \frac{D_m}{D}}^{\text{abiotic reduction}} - \underbrace{\lambda_{D_i} \theta D_m + \lambda_{D_m} D_i}_{\text{MIMT}} \qquad \left[\frac{\text{mol}}{\text{L}_\text{b}\ \text{s}}\right], \tag{12}$$

$$\frac{\partial \theta I}{\partial t} = \mathcal{T}\{I\} \qquad \left[\frac{\text{mol}}{\text{L}_\text{b}\ \text{s}}\right], \tag{13}$$

$$\frac{\partial \theta X}{\partial t} = \mathcal{T}\{X\} - \overbrace{\Gamma_X BX}^{\text{biocide reaction}} \qquad \left[\frac{\text{mol}}{\text{L}_\text{b}\ \text{s}}\right]. \tag{14}$$

### 2.3.3 Partial differential equations for immobile chemical components

The immobile component concentrations are affected only by the reactive processes outlined above:

$$\frac{\partial B}{\partial t} = \overbrace{\mu_B}^{\text{biomass growth}} - \underbrace{\lambda_{B_2}(B - B_{\min})}_{\text{natural decay}} - \overbrace{\Gamma_B (B - B_{\min})X}^{\text{biocide reaction}} \qquad \left[\frac{\text{mol}}{\text{L}_\text{b}\ \text{s}}\right], \tag{15}$$

$$\frac{\partial D_i}{\partial t} = - \overbrace{S_{D_1} \frac{D_i}{D} \mu_B}^{\text{biomass growth}} - \underbrace{\lambda_D \frac{D_i}{D} B}_{\text{respiration}} - \overbrace{S_{D_2} \mu_{CD} \frac{D_i}{D}}^{\text{abiotic reduction}} + \underbrace{\lambda_{D_i} \theta D_m - \lambda_{D_m} D_i}_{\text{MIMT}} \qquad \left[\frac{\text{mol}}{\text{L}_\text{b}\ \text{s}}\right]. \tag{16}$$

## 3 CHROTRAN validation and remediation case studies

The PFLOTRAN software from which CHROTRAN derives its numerical flow and reactive transport solvers has gone through extensive quality assurance testing, has been benchmarked against other reactive transport solvers, and is used inside and outside the U.S. Department of Energy for mission-critical analytical work. The new bio-reactive transport model that is constitutive of CHROTRAN is not available in any other software, so direct benchmarking is not possible. However, extensive quality testing has been performed by the developers. We have validated through batch and multidimensional simulations that CHROTRAN does satisfy the governing equations we present for chemistry and permeability, and also that it gives plausible, physically consistent results for a wide range of scenarios.





To demonstrate the novel capabilities of our software, we present two example studies, which together illustrate the interactions of all the types of chemical species it permits to be modeled, along with its treatment of bio-clogging. The input files for these two examples can be found in the `chrotran_examples` directory of the CHROTRAN repository.

## 3.1 Case study: remediation of Cr(VI) by molasses and ethanol co-injection

This study concerns the co-injection of molasses (electron donor, $D$) and ethanol (nonlethal bio-inhibitor, $I$) into a single well drilled in a heterogeneous aquifer with an appreciable background Cr(VI) concentration. The competition between direct abiotic reduction of Cr(VI) by molasses and bio-reduction of Cr(VI), which exists since both reduction pathways consume the electron donor, along with the impact of suppressing the biomass growth is explored. The basic parameters used are those shown in Figures A1 and A2, with changes as indicated below.

Four related simulations are performed on the same $100 \times 100$ m two-dimensional heterogeneous hydraulic conductivity field, with geometric mean conductivity $K_g = 10^{-4}$ m s$^{-1}$, a multi-Gaussian correlation structure with exponential semivariogram with correlation length of 4 m and $\sigma^2_{\ln K} = 2$. Each simulation takes place over a span of 500 days and begins with $\varepsilon = 10^{-20}$ initial concentrations of all species, except $B(\boldsymbol{x},0) = B_{min} = 10^{-10}$ mol L$_b^{-1}$ and $C = 1.923 \times 10^{-5}$ mol L$^{-1}$ (1000 ppb Cr(VI)). In all cases, there are no flow boundaries at the north and south of the domain ($y = 0$ m and $y = 100$ m), and constant head

boundaries are imposed at the west and east of the domain ($x = 0$ m and $x = 100$ m) such that there is a drop of head of 0.28 m between these faces. A single injection well exists at $(x,y) = (25\,\text{m}, 50\,\text{m})$. For the first 10 days of the simulation, there is no injection into the well. From 10 d to 30 d, injection is performed at the well with constant volumetric flow rate 272.55 m$^3$d$^{-1}$ with species concentrations discussed below. From 30 d to 500 d, there is again no injection at the well. A very large (arbitrary) $\rho_B$ is assumed, so as to eliminate the effect of biomass clogging from this simulation.

The four simulations differ in their chemistry only. Two direct abiotic reduction rates are considered: $\Gamma_{CD} = 1$ L mol$^{-1}$ s$^{-1}$ and $\Gamma_{CD} = 0$ L mol$^{-1}$ s$^{-1}$, as are two different ethanol concentrations in the injection fluid: $I = 1$ mol L$^{-1}$ and $I = \varepsilon$ mol L$^{-1}$, in all four possible combinations. The injection fluid chemistry always has Cr(VI) concentration equal to the initial concentration ($C = 1.923 \times 10^{-5}$ mol L$^{-1}$), ensuring that no chromium disappearance is due to dilution, and molasses concentration $D = 1 \times 10^{-2}$ mol L$^{-1}$.

Concentrations of Cr(VI) for each scenario are shown in Figure 1. It is apparent that little persistent reduction due to biomass alone occurs, although ethanol co-injection does increase biomass footprint, which has a noticeable and persistent effect. By contrast, the rapid abiotic reaction between Cr(VI) and a constituent of molasses has more impact. This is attributable to the fact that molasses has a large reducing capacity, background concentrations of Cr(VI) are relatively low, and it has a retardation factor of around 150 (obtained from Shashidhar et al. (2006)), meaning that it has the potential to form a persistent permeable

reactive barrier around the well. The better performance in the presence of ethanol is attributable to the fact that ethanol co-injection prevented consumption of molasses by the biomass during the injection phase, and so molasses persists over a larger area.





## 3.2   Case study: biomass clogging/unclogging due to acetate/dithionite injection

CHROTRAN has the capability to model hydraulic conductivity reduction due to bio-fouling and the use of biocide as a remediation strategy. To illustrate model capabilities, we perform a simulation of constant-head injection into a homogeneous aquifer in which the injection fluid is amended initially with the biostimulant acetate ($D = 10^{-2}$ M) for the first 400 d. The acetate

amendment is subsequently replaced with the biocide dithionite ($X = 3.5$ M), for the remainder of the simulation. The basic structure of the CHROTRAN input file is the same as in the study outlined in Section 3.1 (this is to say, as shown in Figures A1 and A2), but with different CHROTRAN parameter values, as shown in Table 1. We here make the reasonable (Ritmann, 2004, p. 361) assumption that biomass has the same density as water (recall that we everywhere use the interpretation that 1 mol of biomass is defined as 1 g of biomass).

The simulation is performed on a 50 m square homogeneous hydraulic conductivity field, with constant hydraulic conductivity $K = 9.8 \times 10^{-5}$ m s$^{-1}$. Each simulation takes place over a span of 500 days, and begins with $\varepsilon = 10^{-20}$ initial concentrations of all species, except $B(\boldsymbol{x}, 0) = B_{min} = 10^{-10}$ mol L$_b^{-1}$ and $C = 1.923 \times 10^{-5}$ mol L$^{-1}$. No flow boundaries are imposed at the north and south of the domain ($y = 0$ m and $y = 50$ m), and constant head boundaries are imposed at the west and east of the domain (head 0.28 m at $x = 0$ m and head 0 m at $x = 50$ m). A single injection well exists at $(x, y) = (25$ m, $25$ m), and constant

head of 0.28 m is imposed at its location.

A sequence of quiver plots representing the velocity field at nine points in time, superimposed on the intensity of biomass concentration are shown in Figure 2. During the first 400 d of the simulation, biomass concentration grows in the vicinity of the well, until hydraulic conductivity drops to zero at the well until no influx occurs there; only ambient flow is apparent, flowing around the impermeable barrier near the well. At this point, the biomass has become useless for bioremediation, as

contaminated aquifer water no longer travels through it. However, at 400 d, dithionite is introduced into the injection fluid and effectively eliminates biomass in the vicinity of the well. The region containing dithionite is relatively sterile and grows outwards until the biomass concentration approaches background, and the initial flow regime is recovered at 416 d. Because initial and final conditions are the same, this cycle may be performed indefinitely.

## 4   Summary and conclusions

For modeling in situ remediation of aqueous groundwater contaminants by injection of aqueous amendment, we recognized the importance of mathematical formulations and numerical codes that can represent three-dimensional fluid flow and multi-species contaminant transport in heterogeneous aquifers with arbitrary injection regimes. For the particularly important case of heavy metal remediation, a number of contaminant-remediation processes (pathways) are susceptible to a unified modeling framework: bio-reduction, bio-precipitation, and direct reduction by the chemical amendment. There have previously existed no

general tools appropriate for modeling such interventions. With this background in mind, we developed a mathematical model that describes the reactive transport dynamics of an amendment (containing any combination of electron donor, non-lethal





bio-inhibitor, and biocide) with biomass and aqueous heavy metal contaminant. We also implemented the mathematical model in a novel computational framework, called CHROTRAN, that is based on the open-source code PFLOTRAN. PFLOTRAN's modularity and the reaction sandbox capability allowed us to implement the model easily without making any changes to the flow and transport code of PFLOTRAN. CHROTRAN can harness the high-performance computing capabilities of PFLOTRAN which

allows for simulations of complex models with large number of computational cells and degrees of freedom. We described our computer implementation and explained how to use CHROTRAN to solve practical problems.

We also considered two demonstration studies related to chromium remediation. The presented synthetic problems were formulated to be consistent with real-world groundwater contamination problems and illustrate the capability of CHROTRAN to aid in the engineering design process. In one of the studies, we discovered that, contrary to much existing theory, Cr(VI) reduction

was maximized by injecting molasses and suppressing biomass growth to maximize the direct, abiotic reduction reaction. In the other, we showed the feasibility of pulsed injection of bio-stimulant and biocide to alleviate bio-fouling in the context of ongoing bioremediation.

We observe that because of the abstraction of our model and its parametric flexibility, the CHROTRAN equations can be used to model other reactive transport behaviors besides the heavy metal bio-reduction that we have focused upon, including ba-

15 sic advection-dispersion-reaction interaction (between $C$ and $D$, in the absence of $B$). The bio-reduction model captures any biodegradation that can be represented using a Monod equation, as long as the contaminant represented by $C$ is non-sorbing, and it does not explicitly require the contaminant to be reduced. This potentially allows for modeling the biodegradation of a wide range of organic contaminants, which include but are not limited to hydrocarbons, chlorinated solvents, pesticides, and volatile organic compounds.

*Code availability.* The Fortran source code for CHROTRAN, along with input files for the examples presented in this document, is freely available at https://github.com/scottkalevhansen/CHROTRAN-release, released under the GPL 3 license.

## Appendix A: User manual

### A1 Installing CHROTRAN

CHROTRAN must be compiled using the GFortran complier (freely available as part of the GNU Compiler Collection). It is

25 based on the open-source PFLOTRAN code base, and the installation procedure is essentially the same as that required to build PFLOTRAN from source, and CHROTRAN requires all the libraries upon which PFLOTRAN depends, including PETSc (Balay et al., 2016) and others. For installation of required libraries, the PFLOTRAN installation instructions[1] are applicable, except that

---

[1]Available at http://documentation.pflotran.org/user_guide/how_to/installation/installation.html.





CHROTRAN, rather than PFLOTRAN, should be cloned from its repository[2] once all the dependencies have installed. To build CHROTRAN itself, navigate to `<path of cloned repository>/src/pflotran` and type `make chrotran`. (The CHROTRAN executable will be called `chrotran`.)

## A2 Specifying and running a simulation

A CHROTRAN input file is of the same format as a PFLOTRAN input file. Information on how to set up such a file is available in the PFLOTRAN user manual (Lichtner et al., 2015). However, to use CHROTRAN's additional functionality, a few of the input cards (top-level blocks, in PFLOTRAN jargon) must contain some particular content. The required CHEMISTRY card format is shown in Figure A1, with bold text being mandatory and standard-weight text being user-alterable. The required SIMULATION, MATERIAL_PROPERTY, and (initial) CONSTRAINT card formats are shown in Figure A2, again with bold text being mandatory

and standard-weight text being user-alterable. Comments in the input file are preceded by the character #.

In addition to these cards being properly formatted, there must exist a chemistry database at the (absolute or relative) path specified after the DATABASE keyword in the CHEMISTRY card, and it must, at a minimum contain the lines shown in Figure A3. The one exception to bold text being mandatory is that species *names* can be changed at will, as long as there is consistency between the CHEMISTRY card and the chemistry database. For instance, one could change all instances of the text Cr(VI) in

both of those locations to U(VI) or all instances of the text chubbite to etibbuhc, with no alteration in execution behavior (besides, obviously, the species names used in the output files).

The chemistry database contains lines for five mobile species: water, plus the mobile species in the CHROTRAN kinetics listed in Section 2.3: $C$, $D_m$, $I$, and $X$. The database also contains a line for a "dummy" mineral species, chubbite, which does not correspond to any species previously mentioned. This species is treated as a mineral which is specified as inactive with respect

precipitation/dissolution by setting its kinetic rate constant (RATE_CONSTANT) to zero. The mineral is included as a surrogate for biomass and porous media volume in CHROTRAN and is updated according to Equation 7 to track $1 - \theta(\boldsymbol{x}, t)$. The initial volume fraction of chubbite thus defines the initial porosity. The format of a chemistry database is discussed in more detail in the PFLOTRAN user manual.

Once you have saved your input file as, e.g. `test.in`, it is easy to run the code from the console. Navigate to `<path of`

`cloned repository>/src/pflotran`, and type `chrotran -pflotranin <path to input file>/test.in`. The output of the simulation will be saved in the same directory as the input file. Depending on the options specified in the input file, CHROTRAN can save flow field velocities, concentrations of all species, permeabilities, and porosities at any specified times in an `.h5` format file. This file format can be visualized natively using freely-available standalone tools such as VisIt and ParaView, and are also accessible from Python scripts by means of the `h5py` library and from Julia scripts by means of the

`HDF5` package.

[2]Available at https://github.com/scottkalevhansen/CHROTRAN-release





*Competing interests.* The authors declare no competing interests.

*Acknowledgements.* The authors acknowledge the support of the Los Alamos National Laboratory Environmental Programs.



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





**Figure 1.** Maps of Cr(VI) concentrations [ppb] in the aquifer 470 d after injection ceased in each of the four scenarios discussed in Section 3.1. Injection well location is denoted by a black *X*.



**Figure 2.** A sequence of snapshots of the cell-center groundwater seepage velocity fields and biomass concentration distributions in the example of Section 3.2. Velocity magnitude is indicated by arrow length and direction by arrow orientation; the arrow tails are located at the cell center; biomass concentration $[\mathrm{g\,m^{-3}}]$ is indicated by green intensity in each superimposed map. The initial condition snapshot is shown in the upper left corner, with time increasing in the clockwise direction, until the initial condition is reached again at 416 d. The same scale is used in each snapshot.




```
CHEMISTRY
  PRIMARY_SPECIES
    molasses
    Cr(VI)
    ethanol
    biocide
  END
  IMMOBILE_SPECIES
    biomass
    molasses_im
  END
  MINERALS
    chubbite         # dummy mineral whose volume fraction is 1 - porosity
  END
  REACTION_SANDBOX
    CHROTRAN_PARAMETERS
      NAME_D_MOBILE        molasses
      NAME_D_IMMOBILE      molasses_im
      NAME_C               Cr(VI)
      NAME_B               biomass
      NAME_I               ethanol
      NAME_X               biocide
      NAME_BIOMINERAL      chubbite

      EXPONENT_B           1.0          # alpha [-]

      BACKGROUND_CONC_B    1.e-10       # B_min [mol/L_bulk]

      MASS_ACTION_B        0.d0         # Gamma_B [L/mol/s]
      MASS_ACTION_CD       1.0          # Gamma_CD [L/mol/s]
      MASS_ACTION_X        0.d0         # Gamma_X [L/mol/s]

      RATE_B_1             1.d-5        # lambda_B1 [/s]
      RATE_B_2             1.d-6        # lambda_B2 [/s]
      RATE_C               1.d-10       # lambda_C [/s]
      RATE_D               0.d0         # lambda_D [/s]
      RATE_D_IMMOB         150.d-2      # lambda_D_i [/s]
      RATE_D_MOBIL         1.d-2        # lambda_D_m [/s]

      INHIBITION_B         5.d1         # K_B [mol/L_bulk]
      INHIBITION_C         1.d-7        # K_C [M]
      MONOD_D              1.d-6        # K_F [M]
      INHIBITION_I         1.d-4        # K_A [M]

      DENSITY_B            1.d20        # [mol/L, i.e., g/L]

      STOICHIOMETRIC_C     0.33d0       # S_C [-]
      STOICHIOMETRIC_D_1   1.d0         # S_D_1 [-]
      STOICHIOMETRIC_D_2   0.020833d0   # S_D_2 [-]
    END
  END
  MINERAL_KINETICS
    chubbite
      RATE_CONSTANT 0.d0
    END
  END
  UPDATE_POROSITY
  MINIMUM_POROSITY 1.d-4
  UPDATE_PERMEABILITY
  UPDATE_MINERAL_SURFACE_AREA
  DATABASE ./chem.dat
  OUTPUT
    ALL
    FREE_ION
    TOTAL
  END
  LOG_FORMULATION
END
```

**Figure A1.** Example CHEMISTRY card for CHROTRAN input file. Bold text should not be altered. However, additional species may be added to the PRIMARY_SPECIES, IMMOBILE_SPECIES, MINERALS, and MINERAL_KINETICS blocks, if desired. Additional sandboxes can also be used in the REACTION_SANDBOX block.





```
SIMULATION
  SIMULATION_TYPE SUBSURFACE
  PROCESS_MODELS
    SUBSURFACE_FLOW flow
      MODE RICHARDS
    END
    SUBSURFACE_TRANSPORT transport
      GLOBAL_IMPLICIT
      NUMERICAL_JACOBIAN
    END
  END
END

MATERIAL_PROPERTY soil1
  ID 1
  TORTUOSITY 0.1d0
  PERMEABILITY
    DATASET Permeability
  END
  PERMEABILITY_POWER 1.0
  PERMEABILITY_CRITICAL_POROSITY 0.0
  PERMEABILITY_MIN_SCALE_FACTOR 1.d-4
  CHARACTERISTIC_CURVES cc1
END

CONSTRAINT initial
  CONCENTRATIONS
    molasses    1.d-20 T
    ethanol     1.d-20 T
    biocide     1.d-20 T
    Cr(VI)      1.923d-05 T # 1000 ppb
  END
  IMMOBILE
    biomass     1.d-10  # equal to BACKGROUND_CONC_B
    molasses_im 1.d-20
  END
  MINERALS
    chubbite    0.85 1.0 # 0.85 is initial porosity
  END
```

**Figure A2.** Additional cards that require particular content in order for CHROTRAN to work properly. In the SIMULATION card, the NUMERICAL_JACOBIAN option must be specified. In the MATERIAL_PROPERTY card, the OPTION PERMEABILITY_MIN_SCALE_FACTOR 1.d4 option should be set. In CONSTRAINT cards, species that are not present should have small, but non-zero concentrations assigned. The concentration of NAME_B (biomass, here) should equal BACKGROUND_CONC_B in the CHEMISTRY CARD. Finally, the initial porosity of the system is set by assigning the volume fraction of NAME_BIOMINERAL (chubbite, here). In general, bold text is required. However, other options may be specified, if desired.





```
'temperature points' 8 0. 25. 60. 100. 150. 200. 250.
'H2O' 3.0 0.0 18.0153
'Cr(VI)' 0. 0. 0.
'molasses' 0. 0. 0.
'ethanol' 0. 0. 0.
'biocide' 0. 0. 0.
'null' 0 0 0
'null' 1 0. '0' 0. 0. 0. 0. 0. 0. 0. 0. 0. 0. 0.
'null' 0. 1 1. '0' 0. 0. 0. 0. 0. 0. 0. 0. 0.
'chubbite' 1.0 0   0. 0. 0. 0. 0. 0. 0. 0. 1.0
'null' 0. 1 0. '0' 0. 0. 0. 0. 0. 0. 0. 0. 0.
'null' 1 0. '0' 0. 0. 0. 0. 0. 0. 0. 0. 0. 0. 0.
```

**Figure A3.** Minimal CHROTRAN chemistry database. The text shown here should not be removed, however additional species may be added, if desired. See PFLOTRAN user manual for details on the database format.



**Table 1.** CHROTRAN parameter values used in the bio-fouling example in Section 3.2.

| Symbol | Value | Units |
|---|---|---|
| $\alpha$ | 1 | - |
| $B_{\min}$ | $10^{-10}$ | mol $L_b^{-1}$ |
| $\Gamma_B$ | $2.6 \times 10^{-2}$ | L mol$^{-1}$ s$^{-1}$ |
| $\Gamma_{CD}$ | 1 | L mol$^{-1}$ s$^{-1}$ |
| $\Gamma_X$ | $2.6 \times 10^{-5}$ | L mol$^{-1}$ s$^{-1}$ |
| $\lambda_{B_1}$ | $10^{-5}$ | s$^{-1}$ |
| $\lambda_{B_2}$ | $10^{-15}$ | s$^{-1}$ |
| $\lambda_C$ | $10^{-10}$ | s$^{-1}$ |
| $\lambda_D$ | 0 | s$^{-1}$ |
| $\lambda_{D_i}$ | 1.5 | s$^{-1}$ |
| $\lambda_{D_m}$ | $10^{-2}$ | s$^{-1}$ |
| $K_B$ | $5 \times 10^2$ | mol $L_b^{-1}$ |
| $K_C$ | $10^{-7}$ | M |
| $K_D$ | $10^{-6}$ | M |
| $K_I$ | 1 | M |
| $\rho_B$ | $10^3$ | mol L$^{-1}$ |
| $S_C$ | $3.3 \times 10^{-1}$ | - |
| $S_{D_1}$ | $10^{-5}$ | - |
| $S_{D_2}$ | $2.0833 \times 10^{-2}$ | - |

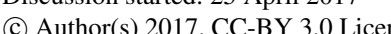



**Table 2.** Relationship between the parameter names in the CHEMISTRY card (Figure A1) and the mathematical symbols shown in Section 2.

| Symbol | Units | Name in CHEMISTRY card |
|---|---|---|
| $\alpha$ | - | EXPONENT_B |
| $B_{\min}$ | mol L$_b^{-1}$ | BACKGROUND_CONC_B |
| $\Gamma_B$ | L mol$^{-1}$ s$^{-1}$ | MASS_ACTION_B |
| $\Gamma_{CD}$ | L mol$^{-1}$ s$^{-1}$ | MASS_ACTION_CD |
| $\Gamma_X$ | L mol$^{-1}$ s$^{-1}$ | MASS_ACTION_X |
| $K_B$ | mol L$_b^{-1}$ | INHIBITION_B |
| $K_C$ | mol L$^{-1}$ | INHIBITION_C |
| $K_D$ | mol L$^{-1}$ | MONOD_D |
| $K_I$ | mol L$^{-1}$ | INHIBITION_I |
| $\lambda_{B_1}$ | s$^{-1}$ | RATE_B_1 |
| $\lambda_{B_2}$ | s$^{-1}$ | RATE_B_2 |
| $\lambda_C$ | s$^{-1}$ | RATE_C |
| $\lambda_D$ | s$^{-1}$ | RATE_D |
| $\lambda_{D_i}$ | s$^{-1}$ | RATE_D_IMMOB |
| $\lambda_{D_m}$ | s$^{-1}$ | RATE_D_MOBIL |
| $\rho_B$ | mol L$^{-1}$ | DENSITY_B |
| $S_C$ | - | STOICHIOMETRIC_C |
| $S_{D_1}$ | - | STOICHIOMETRIC_D_1 |
| $S_{D_2}$ | - | STOICHIOMETRIC_D_2 |