# Peer review of "CHROTRAN 1.0: A mathematical and computational model for in situ heavy metal remediation in heterogeneous aquifers"

_Geoscientific Model Development, 2017_

## Referee Comment (RC1) · M. Walther (Referee) · 21 Jun 2017

Comments on "CHROTRAN 1.0: A mathematical and computational model for in situ heavy metal remediation in heterogeneous aquifers" submitted to GMD by Scott K. Hansen, Sachin Pandey, Satish Karra, and Velimir V. Vesselinov

The manuscript describes the principles of a newly developed numerical model to simulate heavy metal contamination remediation involving biological fate (together with clogging) in saturated porous media; two example setups are provided to show the model's functionality. The approach is based on the open-source simulation toolbox PFLOTRAN and builds upon its functionality by extending the reactive module for the

purpose of simulating the fate of heavy metals.

The structure of the manuscript is logically, providing an initial literature review, followed by a description of the used approach for the model development, before example calculations are presented. The manuscript is written in a clear and concise language; figures are used sparcely, tables summarize required model parameters; an appendix gives a short overview on the file structure for the newly developed features.

My view of the work is twofold. On the one hand, I highly support the publishing of new developments together with a proper benchmarking; this may at the first place not be acknowledged as a scientific advancement, but it very much provides the basis for the latter in many follow-up applications. This is furthermore highlighted by the fact that more and more approaches become increasingly complex which requires a proper documentation for a sustainable development of these approaches. In this light, I strongly support the publishing of this manuscript. On the other hand, I think that there is one major issue with the approach the manuscript presents. Heavy metal remediation is usually governed by the redox potential, which furthermore will change during the reaction and within biomass (biofilm). In the whole manuscript, I could not find any discussion or reasoning why you do not want to consider redox potential in your approach, which is so substantial for the whole reaction system. Having said this, I would like to raise the question whether the approach is sufficient for the broad applicability it claims to have (see page 3, line 30 ff). Also, the two examples only confirm the expected behaviour and cannot be used as benchmarks for the new processes. Therefore, I would like to encourage the authors to give reason 1) why it is ok to neglect redox potential, 2) why their approach is still capable to fulfil the expectations, and 3) provide appropriate benchmarks to ensure correct functioning of the new implementations.

I furthermore listed a number of specific comments below.

I hope that I could help to improve the manuscript and would be available for a second review, if the authors wish so.

[Figure]

**General**

Please sort the variables by their occurence in the equation.

For all variables explained, I suggest to use dimensions (LENGTH, TIME etc), not units (meter, seconds etc), whenever appropriate.

**Introduction**

The introduction nicely lists some published approaches for modeling heavy metal fate in saturated porous media. I understand that your literature study concluded that there was no sufficient model to fulfill the requirements listed in page 4, line 3 (btw: these achievement should rather be given in a summary at the end). I am wondering if you had any incentive to implement such a complex modeling approach (in other words: what was your motivation to develop such a new model)?

Page 3, Line 3: "t" is Time?

Page 3, Line 3: I am wondering, whether the dimensions of the equation are correct (please correct me, if this is a wrong intension). The two left-hand side terms should have [MASS/VOLUME/TIME], but the right-hand side seems to have [MASS/VOLUME]. Can you explain this please?

**Model description**

P5, L11: What is $L_b$?

P5, L11ff: many explained units seem to be concentrations, please state this in the bold names.

P5, L26: - "...is the current porosity at LOCATION x"? - I assume, you define the porosity with $0 < theta < 1$ relative to the total pore volume (ie. 1 - volume of solid phase)? - Why is it not $D = D_i / theta(x,t) + D_m / (1 - theta(x,t))$?

P6, L4ff: q is not defined (you probably can write something like "with the water mass

fluxes related to head via Darcy flux q")

P6, L7ff: - theta is already defined on P5. - please revise the list of variables to use "and" and commas appropriately.

P6, L8: I suggest to move the note on additional capabilities of CHROTRAN to P5, L3, where you already give a small hint on additional features. Here, it seems not necessary to repeat this.

P6, L23: I would like to discuss your statement, that you do not want to include the dispersive flux. I would argue that the distribution of the different components of the reactive system, ie. an aqueous contaminant, an electron donor, a biocide, is majorly governed by advection and hydrodynamic dispersion. With three (or more) governing components, the reactions should especially depend on the mixing of the fluid (and thus the solutes). However, you say that you do not want to consider mixing due to dispersion. Also, in biochemical applications, bacterial growth is usually limited by nutrient availability; growth of biomass often happens at the fringe of the biomass area (not talking about a biofilm here), while the inner areas will have a limited nutrient supply (as outer biomass has already used up all available nutrients). This, again is governed by (transverse) dispersivity and the mixing of the required constituents for biomass growth. Can you please elaborate on this? Besides the discussion on the relevance of dispersion, you sometimes speak of the "advection-dispersion operator" (eg P8, L12), which is somewhat misleading as you do not want to consider dispersion.

P7, L12ff: I like fact that you describe the relevant biochemical processes (L11 says "chemical" processes). The mentioned assumptions, which you often state to be "common" or you assume to follow Monod or linear kinetics, however, should be backed up by a few references who did this in a similar way.

P7, L27: A question of understanding: If biomass decays, e.g. through lysis, will this release the heavy metals again, or will other biomass be able to use this as nutrients?

P9, L1, Eq 11 ff: Maybe, I missed it, but what is S?

P9, L10ff: Please add some references for the testing cases, benchmarks, and applications which PFLOTRAN is used for. Furthermore, I understand that your new implementation may be hard to test against other software that do not have the capability to run these setups. However, I think that you could have chosen examples that have the option to either neglect the new processes or to include them; for the former, you could run alternative models and test them against your implementation; for the latter, differences should be visible that should be validated against mass balances for consistence.

P10, L3: Where can I find the CHROTRAN repository? (Please add a reference to P12, L20.)

P11, L4: What does the unit "M" stand for?

Figure 2: - If t=400d, flow velocities at the well are very small (practically zero?) due to bioclogging. At this point, you start to inject a biocide. I have to questions: 1) As hydr. conductivity is very low, the distribution of the biocide should majorly be governed by diffusion. I am astonished that this relatively large area ($\sim$5x5m$^2$) is remediated so fast. Can you explain this? 2) For all t>400d, the shape of the "remediated area" (where the biocide is injected) shows the shape of a diamond; why isn't this shape similar to the shape of the biomass? Is this a numerical artifact?

**Summary and conclusions**

P11, L26: You did not show the three-dimensional capabilities of your code.

P12, L4: You also did not show the HPC capabilities of your implementation.

**Source Code**

I could find the source code on github, but could not find any pull requests or commits that build upon the original PFLOTRAN code. Therefore, I could not check any of

the new developments you implemented. I highly recommend, especially for further development of your code, to provide a repository that uses as an initial commit an unchanged PFLOTRAN version and then shows your additions as several, logically combined commits.

**Appendix**

P12, L26: Please indicate the PFLOTRAN version you used as a base. Do you think you could easily rebase your code to a future release of PFLOTRAN?

P13, L28: please add reference to VisIt and ParaView. See this for the latter: https://www.paraview.org/publications/

**Typos, Grammar etc**

Page 3, Line 4: Please remove the comma in "where C, is the U(VI) concentration". Later, C is also reused for other heavy metal concentrations, please mention that.

P10, L10: Please change meter to square meter.

P10, L12: epsilon, the initial concentration should have a unit.

P12, L24: "complier" -> "compiler"

---

## Referee Comment (RC2) · Anonymous Referee #2 · 21 Aug 2017

Review of gmd-2017-51

The manuscript "CHROTRAN 1.0: A mathematical and computational model for in situ heavy metal remediation in heterogeneous aquifers" by Hansen et al. (gmd-2017-51) presents a conceptual modeling approach for the reaction-transport simulation of chromium in groundwater. The conceptual approach considering, transport sorption, biotic and abiotic reduction of Cr(VI), growth and decay of microbial biomass, and clogging of the pore space due to biomass accumulation is implemented into the 3D reactive transport environment PFLOTRAN. The performance of the approach is demonstrated using two generic case studies.

[Figure]

The manuscript is well written and the presented approach appears in general technically sound making use of well-established concepts. Some of the assumptions regarding the considered processes and their kinetic description would need a better explanation/justification but my largest concern regarding this manuscript is whether it indeed presents a new model or whether is presents 'just' an application of PFLOTRAN for the simulation of Cr(VI). Given that the shown model applications are two generic scenarios without any in-depth discussion of the results and their potential meaning, it is not possible to validate the applicability of the presented conceptual approach (i.e. set of equations) to real-world scenarios. If – as I appears to me – the novel aspect of the manuscript is restricted to the conceptual approach it would not justify publication of the manuscript.

Specific comments:

P3, L28: Is the only short-coming of the existing models for Cr(VI) reduction the fact that they consider 1D transport only? If so, why is there a need for an alternative description of the reactive processes?

P4, L21: No, there are several other codes which would be capable of simulating the presented processes (perhaps not always the clogging, but certainly all the reactive processes). See e.g., Schäfer et al., 1998, Journal of Contaminant Hydrology 31: 167; Mayer et al., 2002, Water Resources Research 38: 1174; Prommer et al., 2003, Ground Water 41: 247; Centler et al., 2010, Computational Geosciences 36: 397. All these models would be sufficiently flexible to allow describing the presented processes using the set of equations shown further down in the manuscript.

P5,L5-8: While I support this line of approach I am wondering why it would need an 'new' model for its simulation. What is presented in the following is the abiotic and biotic redox transformation of two (partially) mobile species. This is handled by quite a number of reaction-transport models for groundwater settings and it actually does not matter if the electron donor or the electron acceptor is considered as contaminant.

P5, L11: It appears quite strange/confusing introducing B with the unit mol/L but then interpreting 1 mol as 1 g ... Why not stating that the unit of B is up to the user and eventually requires the units of the parameter $S_D$ to be defined consistently.

P5, L26: This implies that the reactivity of the sorbed and the dissolved donor is the same. If this would be the general case, many researchers studying reductions of bioavailability due to sorption would waste their time. Some words of discussion/justification would be needed here.

P6,L12/Eq.6: In the literature one can find a large number of possible relations between changes of porosity and changes of hydraulic conductivity due to (bio)clogging. However, to my knowledge a linear relation has not been proposed, yet. Give reference/justification for this assumption.

P8, L8/Eq.9 and P9, L1/Eq.11: Why is there no dependency of microbial growth on the contaminant/electron acceptor? This implies that everywhere some other (more favorable) electron acceptor must be available at non-limiting concentrations. If this would be the case why should there be a consumption of the heavy metal? Also, why is the bio-reduction rate not controlled by the presence of the electron donor? The equation implies that as long as there is sufficient biomass there would be a bio-reduction activity even if the is no further supply of the electron donor. This does not appear meaningful to me.

P9, L7/Eq. 15: Is there a process-related justification of the existence of $B_{min}$ or has this been introduced for technical/numerical reasons?

P9, L12: No, there are other codes which could be used for this purpose (see comment above for P4, L21). However, I agree that benchmarking is not needed here. PFLOTRAN is well established and any benchmark would not allow determining if the presented concept is meaningful.

P9, L14: Are any of these validations available in the literature? If not this statement

might of course be true but any evidence for this is lacking.

P 10, L8: Clarify, are the parameters shown in Tables 1 and 2 those also shown in Figures A1 and A2. I support showing these figures to visualize how the case specific input has to be provided but for communicating parameter values a table is more appropriate. Also: where do these parameter values come from, literature, own experiments/studies, educated guess or . . . ? What is the initial porosity (especially for the clogging case shown further down)?

P 11, L2: Is a constant head injection a reasonable assumption? Usually wells impose a certain flow rate. As there is no shear force related biomass removal considered I assume that the model would not predict reasonable effects for a fixed injection rate well.

P 11, L20: If the biomass seems to inhibit any injection through the well, the dithionite injection would not lead to any effects as long as the biomass is not decreasing due to natural decay allowing at least some injection to take place. Right?

———————————

---

## Author Comment (AC1) · 19 Sep 2017

**Response to reviewer comments**

In this document, we reprint the comments of both reviewers, interlineated with our responses. Our responses are typeset in indented **_bold italic type_**.

**Comments of M. Walther (referee #1)**

The manuscript describes the principles of a newly developed numerical model to simulate heavy metal contamination remediation involving biological fate (together with clogging) in saturated porous media; two example setups are provided to show the model's functionality. The approach is based on the open-source simulation toolbox PFLOTRAN and builds upon its functionality by extending the reactive module for the purpose of simulating the fate of heavy metals.

The structure of the manuscript is logically, providing an initial literature review, followed by a description of the used approach for the model development, before example calculations are presented. The manuscript is written in a clear and concise language; figures are used sparsely, tables summarize required model parameters; an appendix gives a short overview on the file structure for the newly developed features.

My view of the work is twofold. On the one hand, I highly support the publishing of new developments together with a proper benchmarking; this may at the first place not be acknowledged as a scientific advancement, but it very much provides the basis for the latter in many follow-up applications. This is furthermore highlighted by the fact that more and more approaches become increasingly complex which requires a proper documentation for a sustainable development of these approaches. In this light, I strongly support the publishing of this manuscript. On the other hand, I think that there is one major issue with the approach the manuscript presents. Heavy metal remediation is usually governed by the redox potential, which furthermore will change during the reaction and within biomass (biofilm). In the whole manuscript, I could not find any discussion or reasoning why you do not want to consider redox potential in your approach, which is so substantial for the whole reaction system. Having said this, I would like to raise the question whether the approach is sufficient for the broad applicability it claims to have (see page 3, line 30 ff). Also, the two examples only confirm the expected behaviour and cannot be used as benchmarks for the new processes. Therefore, I would like to encourage the authors to give reason 1) why it is ok to neglect redox potential, 2) why their approach is still capable to fulfil the expectations, and 3) provide appropriate benchmarks to ensure correct functioning of the new implementations.

I furthermore listed a number of specific comments below.

I hope that I could help to improve the manuscript and would be available for a second review, if the authors wish so.

> ***We thank the reviewer for taking the time to provide an objective, thorough review of the submitted manuscript, and for his overall positive comments. We have made a sincere effort to address their comments. With regard to the comments enumerated above:***
>
> 1) ***Although calculation of redox potential helps to identify thermodynamically favorable reactions, the rates at which they will proceed vary significantly due to factors such as reaction overpotential and microbial enzymatic catalysis. In many cases, redox reactions are slow and redox-sensitive species may remain in thermodynamic disequilibrium (Keating and Bahr, 1998). For this reason, mathematical formulations of redox reactions, such as the model presented here, are often assumed to be kinetically limited, and are typically dependent upon the concentration of oxidant and/or reductant as opposed to the redox potential. This is justifiable because the concentration of the oxidant and/or reductant will have to approach exceedingly small concentrations before redox equilibrium is achieved (Steefel and MacQuarrie, 1996). Kinetic-partial equilibrium models have been implemented by others (e.g. McNab and Narsimhan, 1994; Keaton and Bahr, 1998). However, the calculation of redox potential or some other quantity describing electron availability is required. Thus, the redox-potential calculation brings additional parametric uncertainty and the resulting estimates will most likely not agree with field measurements of the electron activity (Eh). Numerous reactive transport models that incorporate biological and abiotic oxidation-reduction reactions for both reactions successfully***

*take a kinetic approach (e.g. Hunter et al., 1998; Mayer et al., 2001, 2002; Li et al., 2009; Molins et al., 2015; Sengor et al., 2015;). Furthermore, this approach is beneficial because it allows the flexibility to calibrate the model using laboratory microcosms or field studies with relative ease. In the current system, it is known that the introduction of an organic carbon source such as molasses will result in reducing conditions, so reduction reactions are the primary driver of remediation. The following text was added to the manuscript (Section 2.2) "The governing equations include kinetically limited redox reactions. These reactions are often non-instantaneous with redox-sensitive species remaining in thermodynamic disequilibrium (Keating and Bahr, 1998), and a kinetic formulation is a fair representation of this type of behavior (Steefel and MacQuarrie, 1996)."*

2) *In light of the above arguments, we believe that a "black box" kinetic approach without explicit treatment of redox potential is most suitable and flexible for the predictive modeling of a variety of heavy metals or other redox-sensitive contaminants for which CHROTRAN is intended.*

3) *We have added a suite of regression tests to the CHROTRAN repository, each of which illustrates the correct functioning of numerical solutions against analytical or empirical benchmarks. The included batch tests cover abiotic reaction, abiotic reaction with sorption (MIMT), microbial growth and decay, as well as interaction with biocide and nonlethal inhibitor. In addition a non-batch reference simulation featuring bio-clogging is included. These benchmarks are located in subdirectories of the* `chrotran_benchmarks` *directory in the developer branch (*`dev`*) of the CHROTRAN repository. In the top-level directory resides a bash script,* `chrotran_benchmarks.sh` *that runs them all. For behavior that is inherited from PFLOTRAN, no specific benchmarks are provided: we concur with the second reviewer that its validity is well established.*

*References for the above citations:*

*Hunter, Kimberley S., Yifeng Wang, and Philippe Van Cappellen. "Kinetic modeling of microbially-driven redox chemistry of subsurface environments: coupling transport, microbial metabolism and geochemistry." Journal of Hydrology 209.1 (1998): 53-80.*

*Keating, Elizabeth Harrison, and Jean M. Bahr. "Reactive transport modeling of redox geochemistry: Approaches to chemical disequilibrium and reaction rate estimation at a site in northern Wisconsin." Water Resources Research 34.12 (1998): 3573-3584.*

*Li, Li, et al. "Mineral transformation and biomass accumulation associated with uranium bioremediation at Rifle, Colorado." Environmental science & technology 43.14 (2009): 5429-5435.*

*Mayer, K. Ulrich, David W. Blowes, and Emil O. Frind. "Reactive transport modeling of an in situ reactive barrier for the treatment of hexavalent chromium and trichloroethylene in groundwater." Water Resources Research 37.12 (2001): 3091-3103.*

*Mayer, K. Ulrich, Emil O. Frind, and David W. Blowes. "Multicomponent reactive transport modeling in variably saturated porous media using a generalized formulation for kinetically controlled reactions." Water Resources Research 38.9 (2002).*

*McNab, W. W., and T. N. Narasimhan. "Modeling reactive transport of organic compounds in groundwater using a partial redox disequilibrium approach." Water Resources Research 30.9 (1994): 2619-2635.*

*Molins, Sergi, et al. "A benchmark for microbially mediated chromium reduction under denitrifying conditions in a biostimulation column experiment." Computational Geosciences 19.3 (2015): 479-496.*

*Şengör, S. Sevinç, et al. "A reactive transport benchmark on modeling biogenic uraninite re-oxidation by Fe (III)-(hydr) oxides." Computational Geosciences 19.3 (2015): 569-583.*

*Steefel, Carl I., and Kerry TB MacQuarrie. "Approaches to modeling of reactive transport in porous media." Reviews in Mineralogy and Geochemistry 34.1 (1996): 85-129.*

**General**

Please sort the variables by their occurrence in the equation.

*We have made an effort to sort all variables as suggested.*

For all variables explained, I suggest to use dimensions (LENGTH, TIME etc), not units (meter, seconds etc), whenever appropriate.

*Although the use of dimensions would allow for a more generalized representation of the mathematical framework, we have elected to maintain the original representation in terms of units because they are consistent with how the equations are represented in the code itself. In particular, there is a hard-coded relationship between the units used for aqueous and bulk concentrations.*

**Introduction**

The introduction nicely lists some published approaches for modeling heavy metal fate in saturated porous media. I understand that your literature study concluded that there was no sufficient model to fulfill the requirements listed in page 4, line 3 (btw: these achievement should rather be given in a summary at the end). I am wondering if you had any incentive to implement such a complex modeling approach (in other words: what was your motivation to develop such a new model)?

*The primary motivation of developing this modeling approach was to simulate the remediation of legacy groundwater contamination at Los Alamos National Laboratory. Since the model is applicable to a wide range of contaminated sites, we have decided not to include our incentive for developing the model.*

Page 3, Line 3: "t" is Time?

*That is correct. This was added to the text.*

Page 3, Line 3: I am wondering, whether the dimensions of the equation are correct (please correct me, if this is a wrong intension). The two left-hand side terms should have [MASS/VOLUME/TIME], but the right-hand side seems to have [MASS/VOLUME]. Can you explain this please?

*Equations (1), (2), and (3) are proportional relationships and do not require consistent units, since they only represent a part of the entire governing equation. In this description, we are illustrating how various Monod expressions used by others influence $\partial C/\partial t$ and decided that writing the entire governing equations would unnecessarily add additional variables. The full governing relationship we use to define $\partial C/\partial t$ is written in Equation (11) and has consistent units [MASS/VOLUME/TIME]. The time dimension on the right-hand side of the expressions arises from the kinetic rate constants $\lambda_C$, $\Gamma_{CD}$, and the advection-dispersion operator.*

**Model description**

P5, L11: What is L_b?

*$L_b$ represents a liter of bulk volume (i.e., total volume of a computational grid cell that includes both the porous medium and the solution), as opposed to a liter of pore water. A brief description of this unit was added to the manuscript.*

P5, L11ff: many explained units seem to be concentrations, please state this in the bold names.

*We believe that having the units written after the bold names as they were in the original submitted manuscript is sufficient. No changes were made.*

P5, L26: - "...is the current porosity at LOCATION x"? - I assume, you define the porosity with 0 < theta < 1 relative to the total pore volume (ie. 1 - volume of solid phase)? - Why is it not D = D_i / theta(x,t) + D_m / (1 - theta(x,t))?

*Dividing $D_i$ by $\theta$ is necessary to express $D_i$ in terms of solution volume as opposed to bulk volume. $D_m$ is already written in terms of solution volume, so division by $1 - \theta(x, t)$ is not required. In the revised manuscript, we have also added a $10^3$ factor to the expression, which converts $m^{-3}$ to $L^{-1}$.*

P6, L4ff: q is not defined (you probably can write something like "with the water mass fluxes related to head via Darcy flux q")

*We have modified the text to clarify this.*

P6, L7ff: - theta is already defined on P5. - please revise the list of variables to use "and" and commas appropriately.

*We have removed the redefinition of $\theta$ and revised the list of variables as suggested.*

P6, L8: I suggest to move the note on additional capabilities of CHROTRAN to P5, L3, where you already give a small hint on additional features. Here, it seems not necessary to repeat this.

*We have taken the reviewers advice and moved this description of additional capabilities to the beginning of the section.*

P6, L23: I would like to discuss your statement, that you do not want to include the dispersive flux. I would argue that the distribution of the different components of the reactive system, ie. an aqueous contaminant, an electron donor, a biocide, is majorly governed by advection and hydrodynamic dispersion. With three (or more) governing components, the reactions should especially depend on the mixing of the fluid (and thus the solutes). However, you say that you do not want to consider mixing due to dispersion. Also, in biochemical applications, bacterial growth is usually limited by nutrient availability; growth of biomass often happens at the fringe of the biomass area (not talking about a biofilm here), while the inner areas will have a limited nutrient supply (as outer biomass has already used up all available nutrients). This, again is governed by (transverse) dispersivity and the mixing of the required constituents for biomass growth. Can you please elaborate on this? Besides the discussion on the relevance of dispersion, you sometimes speak of the "advection-dispersion operator" (eg P8, L12), which is somewhat misleading as you do not want to consider dispersion.

*We agree with the reviewer regarding his scientific remarks about the importance of local-scale dispersion for mixing, and want to stress that CHROTRAN is capable of representing local-scale hydrodynamic dispersion. We did not choose to incorporate it in our examples because the relatively large block size implied non-trivial numerical dispersion and numerical mixing (all Eulerian reactive transport codes implicitly assume that reactants are uniformly distributed in the control volume). We expect the effects of local-scale dispersion to be minor by comparison, and would clutter the exposition. We now also remark explicitly that CHROTRAN does have the capability to handle general dispersion tensors.*

P7, L12ff: I like fact that you describe the relevant biochemical processes (L11 says "chemical" processes). The mentioned assumptions, which you often state to be "common" or you assume to follow Monod or linear kinetics, however, should be backed up by a few references who did this in a similar way.

*We made a sincere effort to describe the most common assumptions and approaches used to model heavy metal bioremediation in the introduction. For this reason, we do not feel the need to cite these works again. However, we did remove the statement "common assumption for bio-mediated processes" from the sentence describing the bio-reduction process.*

P7, L27: A question of understanding: If biomass decays, e.g. through lysis, will this release the heavy metals again, or will other biomass be able to use this as nutrients?

*It is possible that the heavy metal could be assimilated into cells during anabolism. In this case, the heavy metal would be released during lysis. However, enzymatic reduction of the contaminant could also occur extracellularly (e.g. through an electron shuttling process). Our model assumes that reduction is occurring as a dissimilatory reaction that occurs extracellularly, so biomass decay would not necessarily release the heavy metal. The contaminant would have to be re-oxidized prior to its utilization by other biomass.*

P9, L1, Eq 11 ff: Maybe, I missed it, but what is S?

> *The variable S represents stoichiometric relationships between a reaction rate and the consumption rate of a certain species. The variable is defined in the first paragraph of Section 2.2 (Biogeochemical reactions).*

P9, L10ff: Please add some references for the testing cases, benchmarks, and applications which PFLOTRAN is used for. Furthermore, I understand that your new implementation may be hard to test against other software that do not have the capability to run these setups. However, I think that you could have chosen examples that have the option to either neglect the new processes or to include them; for the former, you could run alternative models and test them against your implementation; for the latter, differences should be visible that should be validated against mass balances for consistence.

> *As per the reviewer suggestion, we have added multiple references for PFLOTRAN testing cases, benchmarks, and applications. In order to test our CHROTRAN implementation, we have added additional test problems which are not presented in the manuscript but are available in the developer branch (*`dev`*) of the CHROTRAN repository. We have added the following sentence to the manuscript. "Additional batch test problems can be found in the developer branch of the CHROTRAN repository (*`chrotran_benchmarks` *directory)." These new problems test the various capabilities of CHROTRAN individually in a batch simulator and compare the CHROTRAN numerical solution against analytical or empirical benchmarks implemented in Python. We feel that this additional effort provides compelling evidence that CHROTRAN is working as intended, and it overcomes the difficulty of comparing with other reactive transport models that rely on different conceptual models and mathematical formulations.*

P10, L3: Where can I find the CHROTRAN repository? (Please add a reference to P12, L20.)

> *The CHROTRAN repository is located at* `https://github.com/chrotran/release`*. Additionally, we have setup a CHROTRAN homepage that can be found at* `http://chrotran.lanl.gov`*.*

P11, L4: What does the unit "M" stand for? Figure 2: - If t=400d, flow velocities at the well are very small (practically zero?) due to bioclogging. At this point, you start to inject a biocide. I have to questions: 1) As hydr. conductivity is very low, the distribution of the biocide should majorly be governed by diffusion. I am astonished that this relatively large area (~5x5m2) is remediated so fast. Can you explain this? 2) For all t>400d, the shape of the "remediated area" (where the biocide is injected) shows the shape of a diamond; why isn't this shape similar to the shape of the biomass? Is this a numerical artifact?

> *The unit M represents molar concentration (mol $L^{-1}$). We have changed the units to mol $L^{-1}$ to maintain consistency throughout the text.*

> 1) *The hydraulic conductivity is indeed low enough for flow velocities to be practically zero at t=400 days given the pressure gradients induced by the ambient hydraulic gradient and injection. However, the specified kinetic rate constant, the high concentration of both biomass and biocide, and numerical mixing at the scale of the Eulerian control volumes results in the relatively fast destruction of biomass in the vicinity near the well. Careful inspection of the figure reveals a fringe area (light green) with intermediate biomass concentration around the region that has been completely unclogged (white). Transport in this region is indeed dictated by diffusion and slow advection. However, the combination of factors described above makes the transition from diffusion to advection controlled transport occur at a small enough time scale that the unclogged zone is able to propagate outwards.*

> 2) *The reviewer is correct that this is a numerical artifact: in particular, the "concave diamond" remediated area has the shape that it does on account of numerical dispersion and the relatively coarse grid, whereas in reality should be near-circular. Essentially: numerical dispersion in the $x$ and $y$ directions is proportional to flow velocity through the grid faces that are respectively orthogonal to these directions. If we imagine that the true seepage velocity at the well has magnitude $|v|$, and an effective numerical dispersivity, $\alpha$, applies in both $x$ and $y$ directions, it*

*follows that the effective longitudinal dispersion in those directions is described by Fick's law constant $D = \alpha|v|$. However, for flow at 45 degrees to these axes, it is easy to see that the effective longitudinal dispersion is described by Fick's law constant $D = \frac{\alpha|v|}{\sqrt{2}}$, with dispersion strength in other directions lying between the two extremes. This accounts for the non-physical concave diamond shape near the well. This imperfection can be addressed by increasing the resolution near the well.*

*The reviewer also asked why the shape of the outer bound of the biomass was not the same shape as the interior bound. This is because the flow fields are qualitatively different at these locations. As the interior remediated region begins to develop, it is embedded in a region of thick biomass, and experiences no ambient flow, and so develops in a purely "radial" fashion. By contrast, the exterior of the biomass has ambient flow around its edges, somewhat analogous to the flow around an airplane wing. This flow streamlines the biomass and tends to fill concave regions that we might otherwise see (although note that these would slowly fill due to diffusion, anyhow). In the absence of ambient flow, the exterior of the biomass would have a more pronounced diamond shape, analogous to the remediated region.*

**Summary and conclusions**

P11, L26: You did not show the three-dimensional capabilities of your code.

*We replaced "three-dimensional" with "multi-dimensional" here and in the abstract.*

P12, L4: You also did not show the HPC capabilities of your implementation.

*We replaced "high-performance computing capabilities" with "existing capabilities".*

**Source Code**

I could find the source code on github, but could not find any pull requests or commits that build upon the original PFLOTRAN code. Therefore, I could not check any of the new developments you implemented. I highly recommend, especially for further development of your code, to provide a repository that uses as an initial commit an unchanged PFLOTRAN version and then shows your additions as several, logically combined commits.

*We apologize for the lack of revision history in our original source repository on github. We agree with your suggestion and have updated the source code. The new CHROTRAN repository can be found at* `https://github.com/chrotran/release`*. We have reorganized the repository into a development branch (*`dev`*) and a release branch (*`release`*). By default, release is the default branch. Running* `git log` *in the new repository will show that it maintains all of the change sets found in the* `pflotran-dev` *repository. This enables us to pull and merge changes from* `pflotran-dev` *to both the CHROTRAN release and dev branch and also maintain the simplicity of the original CHROTRAN repository that accompanied the manuscript submission. The latest release version of CHROTRAN possesses the tag* `"v1.0"`*. Minor changes were made to the source code of this version that make it different from the source code submitted with the original manuscript (see the revision log for details). However, these changes were mainly cosmetic and do not change any of the original simulation results presented.*

**Appendix**

P12, L26: Please indicate the PFLOTRAN version you used as a base. Do you think you could easily rebase your code to a future release of PFLOTRAN?

*CHROTRAN version 1.0 in the new GitHub repository is based upon* `pflotran-dev` *commit* `8f33d80`*. This information, along with the required changeset for PETSc (*`03c0fad`*) has been added to the manuscript. Reorganizing the GitHub repository will allow straightforward rebasing of our code to future releases of* `pflotran-dev`*, as described above.*

P13, L28: please add reference to VisIt and ParaView. See this for the latter:
https://www.paraview.org/publications/

*These citations have been added.*

**Typos, Grammar, etc.**

Page 3, Line 4: Please remove the comma in "where C, is the U(VI) concentration".

> *This has been corrected.*

Later, C is also reused for other heavy metal concentrations, please mention that.

> *We have changed the wording to "where C is the heavy metal (U(VI)) concentration".*

P10, L10: Please change meter to square meter.

> *This has been corrected.*

P10, L12: epsilon, the initial concentration should have a unit.

> *We initially did not specify units because the units differ for aqueous and immobile phase concentrations. We have corrected the sentence to include appropriate units for all of the included species. This correction was also made to section 3.2 (biomass clogging case study).*

P12, L24: "complier" -> "compiler"

> *This has been corrected.*

**Comments of anonymous referee #2**

The manuscript "CHROTRAN 1.0: A mathematical and computational model for in situ heavy metal remediation in heterogeneous aquifers" by Hansen et al. (gmd-2017-51) presents a conceptual modeling approach for the reaction-transport simulation of chromium in groundwater. The conceptual approach considering, transport sorption, biotic and abiotic reduction of Cr(VI), growth and decay of microbial biomass, and clogging of the pore space due to biomass accumulation is implemented into the 3D reactive transport environment PFLOTRAN. The performance of the approach is demonstrated using two generic case studies.

The manuscript is well written and the presented approach appears in general technically sound making use of well-established concepts. Some of the assumptions regarding the considered processes and their kinetic description would need a better explanation/justification but my largest concern regarding this manuscript is whether it indeed presents a new model or whether is presents 'just' an application of PFLOTRAN for the simulation of Cr(VI). Given that the shown model applications are two generic scenarios without any in-depth discussion of the results and their potential meaning, it is not possible to validate the applicability of the presented conceptual approach (i.e. set of equations) to real-world scenarios. If – as I appears to me – the novel aspect of the manuscript is restricted to the conceptual approach it would not justify publication of the manuscript.

> *We appreciate the reviewer's assessment of our approach as well written and generally technically sound. We have added some text to the revised manuscript to deal with a number of the reviewer's technical queries.*

> *Regarding the main point of criticism: we stress that it is not correct that CHROTRAN is just an "application" of PFLOTRAN. In its stock form, PFLOTRAN is unable to model either the reaction kinetic equations that we developed, or the effect of bio-clogging. While PFLOTRAN's code is written in a modular way so that those who want to build upon it do not have to reinvent the wheel, substantial software development effort was required in order to build an executable that implements our novel functionality. We estimate that more than 100 man-hours were devoted to software development alone, apart from the conceptual model development and quality assurance, with well over a thousand lines of code added. Thus, the contribution we describe is the development of a new model, its full software implementation, and some example applications. This places our report well inside Geoscientific Model Development's "Model Description Paper" category.*

**Specific comments:**

P3, L28: Is the only short-coming of the existing models for Cr(VI) reduction the fact that they consider 1D transport only? If so, why is there a need for an alternative description of the reactive processes?

> *This is not the only shortcoming relative to our model; we describe in detail how our model dynamics differ from existing bio-reduction models in the introduction. However, it is a major shortcoming of the models we discussed and we feel it is worth stressing.*

P4, L21: No, there are several other codes which would be capable of simulating the presented processes (perhaps not always the clogging, but certainly all the reactive processes). See e.g., Schäfer et al., 1998, Journal of Contaminant Hydrology 31: 167; Mayer et al., 2002, Water Resources Research 38: 1174; Prommer et al., 2003, Ground Water 41: 247; Centler et al., 2010, Computational Geosciences 36: 397. All these models would be sufficiently flexible to allow describing the presented processes using the set of equations shown further down in the manuscript.

> *Even leaving aside the clogging, none of the four papers listed by the reviewer describe capability that is comparable to that which we have developed. Indeed, two of them (Mayer et al. and Prommer et al.) describe models that do not actually treat biomass dynamics explicitly. The other two papers do not include dynamics that we include here. We review all four papers in turn and show how their presented models differ from ours:*

> > 1. *Schäfer et al. (1998) present the model that is closest in spirit to our model, and explicitly treat biomass as a species. They specifically include a first-order mass transfer terms for electron donor and receptor movement between the biofilm and aqueous phase, which we*

*do not. Our model features a more flexible biomass crowding inhibition term and also features the following behavior that was not included in Schäfer et al. (1998):*

    *a. Feedback between biomass growth and permeability.*

    *b. Decay of biomass concentration back to a non-zero background level.*

    *c. Direct reaction between biocide and biomass.*

    *d. Possibility of consumption of electron donor which is proportional to biomass concentration rather than biomass growth.*

2. *Mayer et al. (2002) present a sophisticated flexible reactive transport model for handling aqueous-phase and precipitation-dissolution reactions. However, as the authors themselves write: "[b]acterial growth and die-off is neglected in the present formulation." Microbial reduction can only be treated through use of Monod-type kinetic equations for aqueous species, and biomass concentration and growth rate cannot be factors in these equations.*

3. *Prommer et al. (2003) discuss applications of the PHT3D model. In this paper, an example of bioremediation of chlorinated solvents is modeled, in which every reaction is treated as a first-order decay. Consultation of the PHT3D manual also shows an example in which the Monod and inhibition terms participate in the kinetic equations for aqueous species. No examples were found in which dynamics of immobile biomass concentrations were simulated.*

4. *Centler et al. (2010) present the GeoSysBRNS model, which explicitly treats biomass concentration and discussed a bio-mediated $A + B \rightarrow C$ reaction; The biomass dynamics presented here were simpler than those shown in Schäfer et al. (1998), with first-order decay, linear dependence on biomass concentration, and Monod dependence on the concentrations of A and B. On the evidence of this paper and a 2013 follow-up which is the only other publication listed on the GeoSysBRNS website, the following features of CHROTRAN are not included:*

    *a. Any sort of crowding-based biomass growth inhibition term.*

    *b. Biomass growth inhibition by ethanol or other conservative species.*

    *c. Feedback between biomass growth and permeability.*

    *d. Decay of biomass concentration back to a non-zero background level.*

    *e. Direct reaction between biocide and biomass.*

    *f. Possibility of consumption of electron donor which is proportional to biomass concentration rather than biomass growth.*

*While these papers do not describe models that are equivalent to ours, we have enhanced the literature review section to include the Schäfer et al. (1998) and Centler et al. (2010) papers, and thank the reviewer for this comment.*

P5,L5-8: While I support this line of approach I am wondering why it would need an 'new' model for its simulation. What is presented in the following is the abiotic and biotic redox transformation of two (partially) mobile species. This is handled by quite a number of reaction-transport models for groundwater settings and it actually does not matter if the electron donor or the electron acceptor is considered as contaminant.

*Without reiterating our response to the last question, our model contains a number of bio-reduction-specific features that are not found in other reactive transport models. In particular, we treat biomass explicitly, as an immobile species which occupies space and reduces permeability, which has a background concentration, governed by a hard-limiting biomass crowding term (with tunable exponent), and which can participate in metabolic, abiotic, and conservative inhibition interactions with aqueous species. Some codes may share some of these features, but no existing code has all of them.*

P5, L11: It appears quite strange/confusing introducing B with the unit mol/L but then interpreting 1 mol as 1 g…
Why not stating that the unit of B is up to the user and eventually requires the units of the parameter S_D to be
defined consistently.

> *Our argument for using this formulation is in the paragraph to which the reviewer refers, and we do
> state that the choice of what a "mol" represents is up to the user. We think it is important to show
> the denominator as a bulk volume, which might not be clear if we simply wrote "the choice of the
> unit of B is up to the user". Since some symbol would have to represent amount of biomass in the
> square bracketed unit expression, "mol" seems as good as any.*

P5, L26: This implies that the reactivity of the sorbed and the dissolved donor is the same. If this would be the
general case, many researchers studying reductions of bioavailability due to sorption would waste their time.
Some words of discussion/justification would be needed here.

> *The reviewer is of course correct that the bioavailability of sorbed and aqueous species differ.
> However, as long as the sorbed and aqueous species are in quasi-local-equilibrium, it is possible to
> define an effective reaction rate constant for the total species. We have added text in the paper
> indicating this in response to this comment.*

P6,L12/Eq.6: In the literature one can find a large number of possible relations between changes of porosity and
changes of hydraulic conductivity due to (bio)clogging. However, to my knowledge a linear relation has not been
proposed, yet. Give reference/justification for this assumption.

> *This approximation can be justified by the Zunker empirical formula [e.g., Morin, R.H. "Negative
> correlation between porosity and hydraulic conductivity in sand-and-gravel aquifers at Cape Cod,
> Massachusetts, USA." Journal of Hydrology 316 (2006): 43] which states that $K \propto \frac{\theta}{1-\theta}$. From this, it
> follows immediately that*

$$\frac{K(t)}{K(0)} = \frac{\theta(t)}{\theta(0)}\left[\frac{1 - \theta(0)}{1 - \theta(t)}\right].$$

> *As long as $\theta(0)$ is sufficiently small, the square-bracketed term may be treated as unity, and the
> fractional reduction in permeability equal to the fractional reduction in porosity. We show below a
> numerical example in which an initial 25% porosity is reduced to zero, with the near linearity of the
> corresponding K as a function of $\theta$:*

[Figure]

P8, L8/Eq.9 and P9, L1/Eq.11: Why is there no dependency of microbial growth on the contaminant/electron
acceptor? This implies that everywhere some other (more favorable) electron acceptor must be available at non-
limiting concentrations. If this would be the case why should there be a consumption of the heavy metal? Also,
why is the bio-reduction rate not controlled by the presence of the electron donor? The equation implies that as
long as there is sufficient biomass there would be a bioreduction activity even if the is no further supply of the
electron donor. This does not appear meaningful to me.

*We discussed our decision to neglect dependence on the electron acceptor in lines 21-29 on p. 3 of the original manuscript, and included references to modeling and experimental precedent. To elaborate: numerous processes may be involved in microbial heavy metal reduction: usage of the heavy metal as the terminal electron acceptor (TEA) in cellular respiration, incidental enzymatic reduction, and abiotic reduction by metabolites [Dhal, B., et al. "Chemical and microbial remediation of hexavalent chromium from contaminated soil and mining/metallurgical solid waste: a review." Journal of Hazardous Materials 150-151 (2013): 272]. Bio-reduction of Cr(VI) has been observed even under aerobic conditions in the subsurface [ibid], so it is overly restrictive to assume that the heavy metal will be restricted to the role of the TEA. We use a general approach to modeling the consumption of the heavy metal by the biomass, in which its rate of microbial consumption may be treated as an arbitrary linear combination of the biomass growth rate and biomass concentration. We believe that this should cover most field scenarios, except perhaps for some very specific experiments in which there is no TEA availability besides the single metal contaminant.*

*It has also been shown in batch experiments that cells which are grown with an electron donor, washed, and then placed in suspension with Cr(VI) and no energy source are still able to reduce the Cr(VI) to Cr(III). This so-called "endogenous respiration" has been described for a variety of metal-reducing scenarios [Fredrickson, J. K., H. M. Kostandarithes, S. W. Li, A. E. Plymale, and M. J. Daly. "Reduction of Fe(III), Cr(VI), U(VI), and Tc(VII) by Deinococcus radiodurans R1." Applied and Environmental Microbiology 66 (2000): 2006].*

P9, L7/Eq. 15: Is there a process-related justification of the existence of B_min or has this been introduced for technical/numerical reasons?

*$B_{min}$ represents the background concentration of indigenous biomass that exists in the aquifer in the absence of bio-stimulation. Including $B_{min}$ does provide a numerical benefit in that it prevents B from reaching exceedingly small values when no amendment is present, which could cause transport solver convergence issues. Exceedingly small concentrations of B could also prevent growth during the addition of an amendment, since the growth rate ($\mu_B$) is a function of B.*

P9, L12: No, there are other codes which could be used for this purpose (see comment above for P4, L21). However, I agree that benchmarking is not needed here. PFLOTRAN is well established and any benchmark would not allow determining if the presented concept is meaningful.

*We reiterate that it is not true that other codes could be used for our governing equations, but agree that the reliability of the numerical solvers and I/O handling that we borrow from PFLOTRAN is well-established.*

P9, L14: Are any of these validations available in the literature? If not this statement might of course be true but any evidence for this is lacking.

*Nothing is currently available in the literature, as this is the first manuscript written regarding CHROTRAN. In light of both reviewers' comments regarding CHROTRAN benchmarking, we have now included regression test routines in the repository, which are accessible to all. We have added the following sentence to the manuscript. "Additional batch test problems can be found in the* dev *branch of the CHROTRAN repository (in the* chrotran_benchmarks *directory)." These new problems test the various capabilities of CHROTRAN individually in a batch simulator and compare the CHROTRAN numerical solution against analytical or empirical benchmarks implemented in Python. We feel that this additional effort provides compelling evidence that CHROTRAN is working as intended.*

P 10, L8: Clarify, are the parameters shown in Tables 1 and 2 those also shown in Figures A1 and A2. I support showing these figures to visualize how the case specific input has to be provided but for communicating parameter values a table is more appropriate. Also: where do these parameter values come from, literature, own experiments/studies, educated guess or…? What is the initial porosity (especially for the clogging case shown further down)?

*We indicate that the exact input files used for both of our examples are available in the repository, and so to avoid redundancy we did not duplicate them in the manuscript: Figures A1-A3 are provided as user manual examples, only. In these examples, parameters are chosen for convenience and illustrative value. Although they are intended to be realistic, they are not based on any particular site or set of experiments. We agree that failing to indicate the initial porosity used for the clogging example was an oversight and we have corrected this.*

P 11, L2: Is a constant head injection a reasonable assumption? Usually wells impose a certain flow rate. As there is no shear force related biomass removal considered I assume that the model would not predict reasonable effects for a fixed injection rate well.

*CHROTRAN is capable of handling essentially arbitrary conditions at wells, including constant and time-variant head and constant and time-variant flow rates. We chose a constant-head boundary because it best illustrates the change in flow rate around the well due to bio-clogging. An imposed constant flow rate would (a) show no change in flow regime, and (b) be non-physical, since any pump has only finite ability to create a pressure differential, which would be eventually overcome as the permeability around the well dropped to zero.*

P 11, L20: If the biomass seems to inhibit any injection through the well, the dithionite injection would not lead to any effects as long as the biomass is not decreasing due to natural decay allowing at least some injection to take place. Right?

*From an engineering point of view, there is always a contact front between dithionite-containing pore water and biomass (and indeed some slow, irregular flow). So, it is not required in actuality for biomass to begin dying on its own for dithionite to be effective. From a numerical point of view, numerical mixing at the scale of the Eulerian control volumes ensures some "contact" between biomass and biocide.*